# Congenital Heart Disease: Growth Evaluation and Sport Activity in a Paediatric Population

**DOI:** 10.3390/children9060884

**Published:** 2022-06-14

**Authors:** Thomas Zoller, Maria Antonia Prioli, Maria Clemente, Mara Pilati, Camilla Sandrini, Giovanni Battista Luciani, Marco Deganello Saccomani, Benjamim Ficial, Marcella Gaffuri, Giorgio Piacentini, Lucia Calciano, Angelo Pietrobelli

**Affiliations:** 1Paediatric Clinic, Department of Surgery, Dentistry, Paediatrics and Gynaecology, University of Verona, 37126 Verona, Italy; thomaszoller93@gmail.com (T.Z.); marco.deganellosaccomani@aovr.veneto.it (M.D.S.); benjamim.ficial@aovr.veneto.it (B.F.); marcella.gaffuri@aovr.veneto.it (M.G.); giorgio.piacentini@univr.it (G.P.); 2Division of Cardiac Surgery, Department of Surgery, Dentistry, Paediatrics and Gynaecology, University of Verona, 37126 Verona, Italy; maria.prioli@aovr.veneto.it (M.A.P.); mara.pilati@aovr.veneto.it (M.P.); camilla.sandrini@aovr.veneto.it (C.S.); giovanni.luciani@univr.it (G.B.L.); 3Unit of Epidemiology and Medical Statistics, Department of Diagnostics and Public Health, University of Verona, 37126 Verona, Italy; lucia.calciano@univr.it; 4Pennington Biomedical Research Center, Baton Rouge, LA 70808, USA; angelo.pietrobelli@univr.it

**Keywords:** paediatric cardiology, congenital heart disease, obesity, sports activity

## Abstract

(1) Objective: To evaluate: (i) the associations of age and disease severity with anthropometric indices and weight status, (ii) the difference in the frequency of sports activity among different levels of disease severity in paediatric patients with congenital heart disease (CHD). (2) Methods: Clinical data of Caucasian children (aged 2–18 years) diagnosed with CHD (2005–2018) were retrospectively collected from the electronic register of outpatient visits. Of the 475 children with CHD, 368 children and their 1690 complete anthropometric measurements were eligible for inclusion in our analysis. (3) Results: Significant increase with age was observed for weight z-score [beta (95%CI): 0.03 (0.02, 0.05) for one-unit of age] and BMI z-score [0.06 (0.03, 0.08)] but not for height z-score. The probability of being underweight and overweight/obese increased and decreased with disease severity, respectively. The obesity probability of patients with mild CHD (0.06 [95%CI: 0.03, 0.08]) was not statistically distinguishable from that of patients with moderate CHD (0.03 [95%CI: 0.02, 0.05]), whereas it was lower in patients with severe CHD (0.004 [95%CI: 0.0, 0.009]). No obese patients with a univentricular heart defect were observed. Days spent in sport activities were equal to 1.9 [95%CI: 1.6, 2.2] days/week, 1.9 [1.5, 2.2], 1.4 [1.1, 1.7] and 0.7 [0.1, 1.3] in patients with mild, moderate, severe and univentricular CHD, respectively. (4) Conclusions: The risk of being overweight and obese should not be underestimated in paediatric patients diagnosed with CHD, especially in children with mild or moderate heart defects. It could be prevented or reduced by promoting a healthy lifestyle.

## 1. Introduction

Congenital heart disease (CHD) is the most common congenital disorder [1] and may present with a broad and heterogeneous spectrum of signs and symptoms depending on the cardiac defect [2]. Due to significant advances in paediatric cardiology and cardiac surgery, an increasing number of children with CHD are reaching adulthood [3]. Growth delay remains one of the most common medical issues faced by these children [4]. However, as an increasing number of individuals with CHD reach adulthood, acquired cardiovascular risk factors such as hypertension, smoking, type 2 diabetes, dyslipidaemia, and obesity are becoming increasingly important. These conditions are well known and can be found in adults with CHD [5]. In the adult population with CHD, the main cause of death continues to be heart-failure-related CHD [6], although comorbidities and additional cardiovascular risk factors still play a key role as determinants in mortality, especially in people who reach elderly ages [7]. The prevalence of obesity has been gradually increasing in children and adolescents worldwide [8]. Recent studies have confirmed a similar trend also in the CHD population [9,10]. The recommendation to limit physical activity along with the consumption of high-calorie foods to compensate for slow growth are factors that may increase the risk of being overweight and obese in children with CHD [9,10]. Considering that physical inactivity and obesity in adults with CHD may exacerbate cardiovascular risks, measures to control excessive weight gain are a key part of the care of patients with CHD. The present retrospective longitudinal study aimed to evaluate the associations of age and CHD severity with anthropometric indices and weight status in Caucasian children diagnosed with CHD in Verona, (Italy). In addition, the secondary aim of the current study was to evaluate the difference in the frequency of sports activity among different levels of CHD severity [11].

## 2. Materials and Methods

### 2.1. Study Design

This is a retrospective review of prospectively collected data. Clinical data of children ranging from 2 to 18 years of age, diagnosed with CHD from December 2005 to October 2018, were retrospectively collected from the electronic register of outpatient visits of the Paediatric Cardiology Unit of the Women and Children’s University Hospital of Verona (Italy).

For each patient, the following parameters were recorded during each clinical visit: sex, date of birth, ethnicity, height, weight, body mass index (BMI), disease severity, surgical procedures, and comorbidities. Information on patients’ sports activity was collected through a single telephone interview with their parents or guardians at the end of the study (October 2018).

All methods were carried out in accordance with relevant guidelines and regulations.

### 2.2. Study Participants

Of the 475 Caucasian children (aged 2–18 years) diagnosed with CHD, 107 who had comorbidities that could affect their growth and development (Appendix A) or who had only one measurement information on either weight or height were excluded from the study (Figure 1).

In addition, a total of 24 (1.4%) duplicated medical records (based on identification code, birthdate, and measurement date) and measurements showing height loss over time were excluded from the analysis. Therefore, 368 children and their 1690 complete anthropometric measurements (median [range]: 7 [1,2,3,4,5,6,7,8,9,10,11,12,13,14,15,16,17,18,19,20,21] per patient) were eligible for inclusion in our analysis (Figure 1). Of these, 266 patients were reachable by telephone at the end of the study.

### 2.3. Definitions

The anthropometric indices measured included height (m), weight (kg), and body mass index (BMI, kg/m^2^). Weight, height, and BMI were converted into age- and sex-based z-scores based on the lambda, mu, sigma (LMS) method using the following formula: Z = ((X/M)L − 1)/(L*S). LMS values were obtained from the Cacciari standards for the central-northern Italian population [12].

The cut-off points for being overweight and obese were set at the 75th and 95th percentiles of BMI-for-age and sex, respectively. Patients with a BMI below the 5th percentile were classified as underweight.

At the end of the study, CHD severity was classified as mild (biventricular without any history of surgical intervention), moderate (biventricular with simple defects and a history of surgical interventions), severe (biventricular with complex defects and a history of surgical interventions), univentricular (history of single ventricle diagnoses or palliative surgery including Norwood, Glenn, and Fontan) and unclassified. Simple and complex defects were defined according to the hierarchy diagnosis proposed by Erikssen et al. [13].

Frequency of sports activity was reported by parents/guardians at the telephone interview (“How many days a week does your child exercise/participate in sport activities?”) at the end of the study.

### 2.4. Statistical Analysis

To assess the associations of age and CHD severity with (i) anthropometric indices and (ii) weight status (normal weight, underweight, overweight, obesity), two-level linear and multinomial logistic regression models [14] were used, respectively. Two-level models accounted for the correlation between repeated measures on the same individual: the first level represents the repeated measures clustered within individuals (level 1 units), and the second level is the individual level (level 2 units).

The two-level linear regression model had weight, height, or BMI z-score as outcome, a random intercept term at level 2, a random slope for age at level 2, an unstructured variance–covariance matrix of the random effects at level 2, a 1st order autoregressive error at level 1, and sex and age at clinical visit as fixed effects. The two-level multinomial logistic regression model had weight status as outcome, a random intercept term at level 2, an unstructured variance–covariance matrix of the random effects at level 2, and sex and age at clinical visit as fixed effects. Both two-level regression models were also estimated according to CHD severity, adding the disease severity variable and the indicator of surgery/intervention at the time of measurement (yes/no) to the fixed part of the model. 

In the two-level linear regression model, the beta regression coefficient represents the degree of change in the z-scores for each one-unit difference in the covariate if the other covariates remain constant. In the two-level multinomial logistic regression model, the strength of the associations was measured by the relative risk ratio (RRR), which indicates how the risk of the outcome falling in the comparison group (underweight/overweight/obese patients) compared to the risk of the outcome falling in the referent group (normal weight patients) changes with age or disease severity. The two-level multinomial logistic regression model also allows estimating the probability of being a normal weight/underweight/overweight/obese patient with respect to age and disease severity.

In addition, a Poisson regression model was used to assess the difference in the frequency of sports activity among different levels of CHD severity. The model had the number of days spent per week in sports activities as the outcome, CHD severity as the independent variable, and the age at the time of the phone interview and sex as the adjustment variables.

In all models, age was included as a linear covariate since non-linear terms did not improve goodness-of-fit. Furthermore, the interaction terms between severity and age could not be evaluated due to sparseness of data.

The statistical analyses were carried out using STATA 15 (StataCorp, College Station, TX, USA).

## 3. Results

### 3.1. Patient Characteristics

Of the 368 patients (40.8% female) included in our analysis, the median age at first recording was 5 years of age (range: 2 to 14 years). The median follow-up was 5.2 years (range: 0.0–12.7 years), and 24.7% of our patients had only one recording of height and weight in the medical register. The distribution of CHD severity was: 143 (39.9%) mild, 97 (26.3%) moderate, 68 (18.5%) severe, 12 (3.3%) univentricular and 48 (13.0%) unclassified. At the time of the first recording, most of the children had normal weight (58.2%), 24.7% were underweight and 17.1% were overweight or obese patients. On average, the 266 children (aged 7–18 years) who were reachable by phone spent less than two days a week doing sports activities in the last year (Table 1).

### 3.2. Anthropometric Indices

In the regression model adjusted only for sex (Model 1 in Table 2, Figure 2), an increase with age was observed for both weight z-score [beta regression coefficient (95%CI): 0.03 (0.02, 0.05) for one-unit of age] and BMI z-score [0.06 (0.03, 0.08)] but not for height z-score. The strength of these associations did not change when the severity variable was added to the models (Model 2 in Table 2). Levels of weight and BMI z-scores decreased with increasing severity of disease, whereas height z-score decreased in a statistically significant matter only in children with a univentricular heart defect as compared to patients with mild CHD.

### 3.3. Weight Status

The expected probability of being underweight, overweight, and obese as a child diagnosed with CHD was 0.20 [95%CI: 0.17, 0.23], 0.14 [0.12, 0.16], and 0.03 [0.02, 0.04], respectively, regardless of age and disease severity. The risk of being underweight decreased with increasing age [relative risk ratio (RRR) (95%CI): 0.89 (0.84, 0.93) for 1 year], whereas the risk of obesity increased [1.27 (1.09, 1.49)] and that of overweight remained stable [1.03 (0.98, 1.08)] (Figure 3). Furthermore, the probability of being underweight increased with disease severity, and the probability of being overweight and obese consequently decreased (Figure 4). In fact, the probability of being underweight was statistically significantly higher in patients with univentricular heart defect (0.37 [95%CI: 0.20, 0.55]) and severe CHD (0.25 [95%CI: 0.19, 0.31]), as compared to patients with mild CHD (0.12 [95%CI: 0.9, 0.16]). Instead, it was statistically indistinguishable between patients with moderate (0.19 [95%CI: 0.14, 0.23]) and those with mild CHD. In addition, we found no difference in overweight probability between patients with mild (0.20 [95%CI: 0.15, 0.25]), moderate (0.16 [95%CI: 0.12, 0.20]) and severe (0.11 [95%CI: 0.07, 0.15]) CHD, but the probability was lower in patients with univentricular heart defect (0.01 [95%CI: 0.00, 0.03]). Finally, the obesity probability of patients with mild CHD (0.06 [95%CI: 0.03, 0.08]) was not statistically significant compared to that of patients with moderate CHD (0.03 [95%CI: 0.02, 0.05]). It was lower in patients with severe CHD (0.004 [95%CI: 0.0, 0.009]) compared to patients with mild CHD. No obese patients with a univentricular heart defect were observed.

### 3.4. Sport Activity

Among the 266 patients who were reachable by phone (72.3% of the total), patients with moderate or severe CHD had an expected number of days spent in sports activities (mean [95%CI]: 1.9 [1.5, 2.2] days/week and 1.4 [1.1, 1.7], respectively), that was not statistically different from those with mild CHD (1.9 [1.6, 2.2]). On the other hand, patients with univentricular heart defects reported spending fewer days in sports activities (0.7 [0.1, 1.3]) than patients with mild or moderate CHD (Figure 5).

## 4. Discussion

In the present study, we evaluated the associations of age and CHD severity with anthropometric indices and weight status in Caucasian children diagnosed with CHD, using data retrospectively collected from the hospital register in Verona, Italy. In addition, we evaluated the difference in the frequency of sports activity among different levels of CHD severity.

### 4.1. Anthropometric Indices

In order to obtain weight and BMI z-scores, Italian growth curves [12], were used, these are similar to those of the WHO but are more representative of the country where these patients live. It was found that weight and BMI z-scores increased with increasing age in our paediatric patients with CHD, whereas height z-score did not change. These results are in line with a previous study that suggests a trend toward increasing adiposity over time [15]. In addition, an inverse relationship between anthropometric indicators and CHD severity was observed. In fact, patients with mild CHD had statistically significantly lower BMI and weight z-scores compared to patients with moderate CHD or a univentricular heart defect. This supports a previous study that pointed out that cardiac surgery, when performed during early childhood, could have a remarkable impact on physical growth [16].

### 4.2. Weight Status

The prevalence of obesity has been steadily increasing over the past decades [8] and the trend is similar in patients with CHD [9,10,15]. Recent studies using the American dataset [9] and the Canadian CHD registry [17] found no difference in overweight and obesity rates between CHD and non-CHD paediatric populations. In a recent cohort study, the rates of metabolic syndrome were similar between adults with CHD and healthy control subjects [18].

As reported in previous studies, it was found that the probability of being underweight decreased substantially with age in paediatric patients with CHD [10,16], but it was lower than the probability of being overweight/obese from 10 years of age onwards. This BMI shift from underweight to overweight and obesity was firstly reported in Asian children and adolescent patients [19]. The high probability of being underweight in our patients with the most severe disease state is likely due to chronic long-term issues such as heart failure and failure to thrive that often affect patients with univentricular heart defects [20]. In fact, various factors contribute to the impaired growth and lower body mass index (BMI) in this category of patients: decreased energy intake, malabsorption, increased basal energy requirements, and impaired haemodynamic status [20]. Seventeen-point one percent of our sample was overweight or obese, whereas the overall prevalence of obesity is 17.0% and the overall prevalence of being overweight, including obesity, is 39.4% in the general population in Italy [21]. However, our advice is to strictly monitor the development of these risk factors. In fact, being overweight and obese at the paediatric age increase the risk of morbidity/mortality and comorbidities in patients with heart disease, particularly in cases of cardiac surgery or other interventions [15,22].

### 4.3. Sport Activity

A recent review by Caterini et al. reports that up to 38% of patients experience some level of restriction by their cardiologists, and 70% of caregivers report that their child’s activity is restricted [23]. However, most patients with CHD are relatively sedentary even when they are not restricted by their physiology or when no exercise restrictions have been imposed by their cardiologists [20]. 

The repercussions of a sedentary lifestyle result in an increased risk of being overweight and obese and related comorbidities, even after cardiac repair [20]. In adults with CHD, low exercise capacity is associated with an increased risk of hospitalisation and death [23].

In our study, it was chosen to investigate children’s sport and exercise activity rather than their normal daily physical activity because of the technical difficulties in assessing how much a child physically moves during the week. Although physical activity and practicing a sport are not the same, they have proven to be closely related [24]. Indeed, patients who exercise more are usually more active in general and less prone to lead a sedentary lifestyle.

In line with previous studies [24], our patients reported low levels of daily sports activity, which decreased in patients with more severe cardiac defects. In fact, patients with a biventricular complex defect and a history of surgical interventions spent less than 1.5 days per week in sports activities, although tetralogy of Fallot and transposition of the great arteries defects do not result in any limitations to moderate activity [25,26]. Patients with a univentricular heart defect, (e.g., those who had been palliated with a Glenn and/or Fontan procedure) were physically active less than one day per week. The attitude towards sports activity of children with moderate CHD did not differ from that of children with mild severity—about 2 days of activity per week. These results are daunting considering the physical and psychosocial health benefits of an active lifestyle. Active children with CHD develop better motor capacity [27], have a higher quality of life [28], greater self-esteem [29], and have a lower risk of being overweight or obese [30].

Accordingly, the American Heart Association has encouraged daily participation in appropriate physical activity for all patients with CHD, with the exception of patients with rhythm disorders and the avoidance of contact sport in patients with anticoagulation therapy [31]. At least 60 min of physical activity per day is recommended for children; vigorous activities are recommended at least 3 days a week [31]. Furthermore, the 36th Bethesda Conference and the Association of European Pediatric Cardiology asserted that children with a univentricular heart defect, after a complete diagnostic evaluation, can participate in low-intensity competitive sports like bowling and golf and, if the ventricular function lies within normal range, sports like table tennis and volleyball [25,26]. In Italy, the Cardiological Organizing Committee for Sport Suitability allows moderate to vigorous physical activities after an extensive cardiological evaluation. The latter is individualised according to the severity of CHD and its correction and can include cardiac ultrasound, ECG, and even cardiopulmonary exercise testing and cardiac MRI in severe cases [32]. At our institution, we followed the latter instructions. However, in spite of all these recommendations, practitioners are still often reluctant to encourage an active lifestyle for children with CHD because they may not have extensive knowledge about cardiac effects and risks related to sports activity [33]. Furthermore, physical activity restrictions may sometimes be self-imposed or initiated by anxious parents [24] due to conflicting or incomplete information received from practitioners [33]. In addition, socio-cultural factors can affect physical activity levels. In fact, bullying by peers and a lack of understanding by teachers are not uncommon for children with CHD [33].

According to the literature [19,34], given the low levels of sports practice observed in our population, this study supports the need for the implementation of physical activity and exercise in common clinical practice, for instance by approaching exercise prescription the same way medical therapy is approached. 

## 5. Study Limitations

This is a retrospective longitudinal study on medical records. Therefore, the data collection was not standardised, nor were measurements filed at regular time intervals. Of note, 24.7% of our subjects had only one recording of anthropometric data. In order to assess the selection bias due to the inclusion of these subjects, first, we compared the main characteristics at the visit between patients with only one clinical visit (S1) and patients with at least two clinical visits (S2). We found that S1 patients were younger and had a lower level of disease severity than S2 patients, as shown in Table 3. Then, we checked whether our results changed when S1 subjects were excluded from the analysis. We found that our results did not change from those in the main analysis. Therefore, we considered our results robust to this selection bias.

Moreover, we did not perform an objective and longitudinal assessment of physical activity, because, in the routine follow-up of these categories of patients, the latter was not included. For this reason, we decided to collect information from guardians/parents only on the frequency of sports activity. We acknowledged that the latter may not be accurate, compared to objective measures of physical activity. However, although physical activity and practicing a sport are not the same, they have proven to be closely related [23]. Indeed, patients who are practicing sport more often are usually less prone to lead a sedentary lifestyle.

In addition, information on medication use and socioeconomic status, which may influence body composition, was not available. 

Finally, in interpreting our results, it should be noted that the patients studied were from a single centre.

## 6. Conclusions

Being underweight remains one of the main problems in paediatric patients diagnosed with CHD, especially at a young age; but the risk of being overweight and obese should not be understated. In fact, patients tend to increase adiposity with increasing age, most likely because their physical activity levels are lower than those of national and international recommendations. Most of our patients with mild and moderate defects were overweight or obese, despite their participation in appropriate physical activity as recommended. Therefore, our study emphasises that promoting a healthy lifestyle is necessary to prevent the occurrence of obesity and being overweight, especially in children with mild or moderate heart defects. These findings should be taken into account when planning and setting up counselling programs to promote better nutrition and physical activity for paediatric CHD patients and their families.

## Figures and Tables

**Figure 1 children-09-00884-f001:**
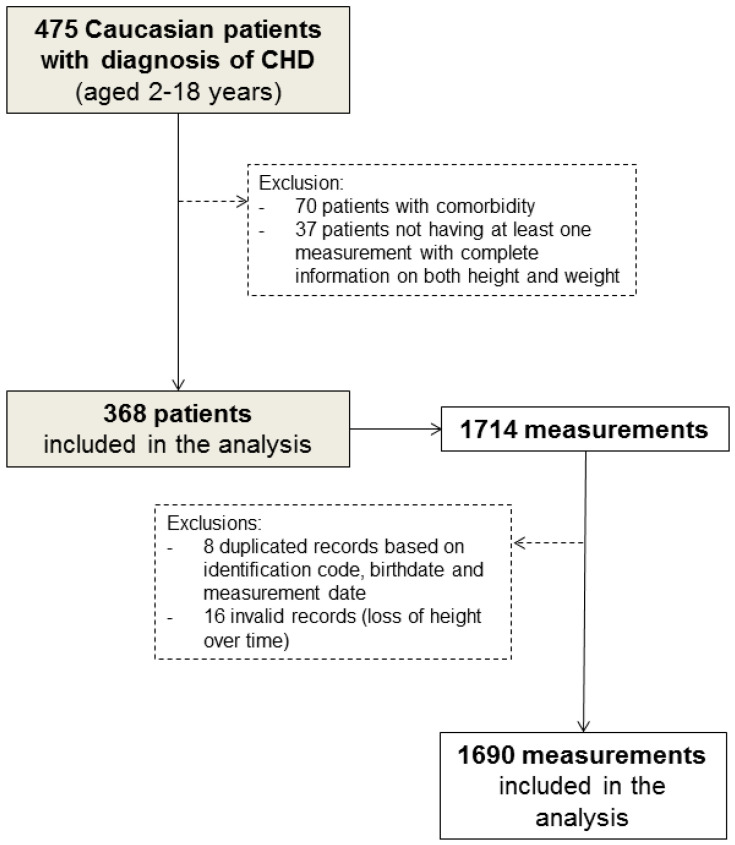
Flow chart of patient selection.

**Figure 2 children-09-00884-f002:**
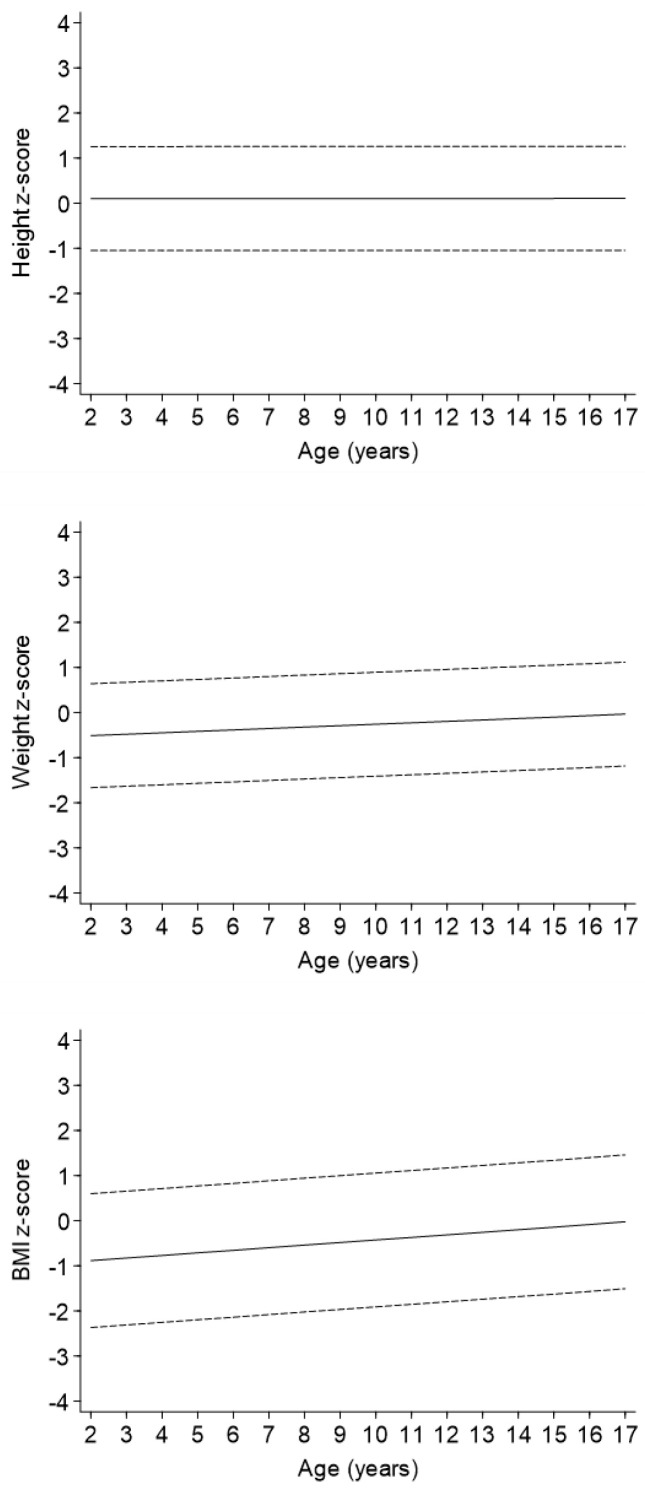
Change in height, weight, and BMI z-scores levels with age.

**Figure 3 children-09-00884-f003:**
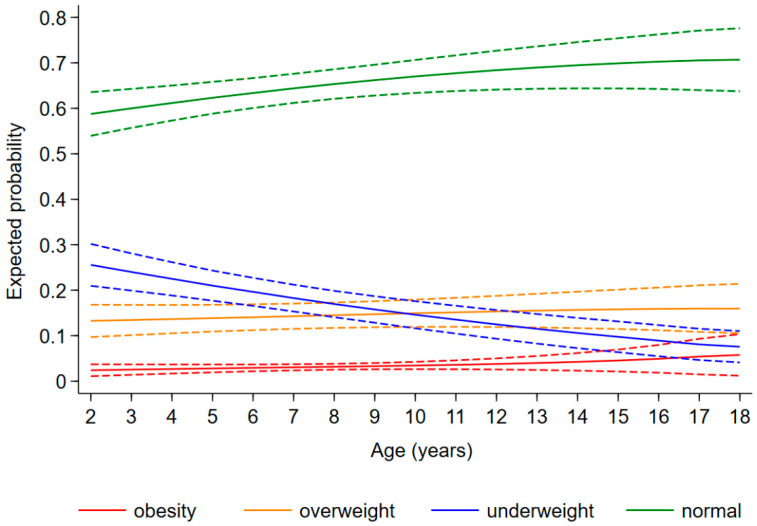
Change in expected probability of underweight (blue), normal weight (green), overweight (orange), and obesity (red) children with age.

**Figure 4 children-09-00884-f004:**
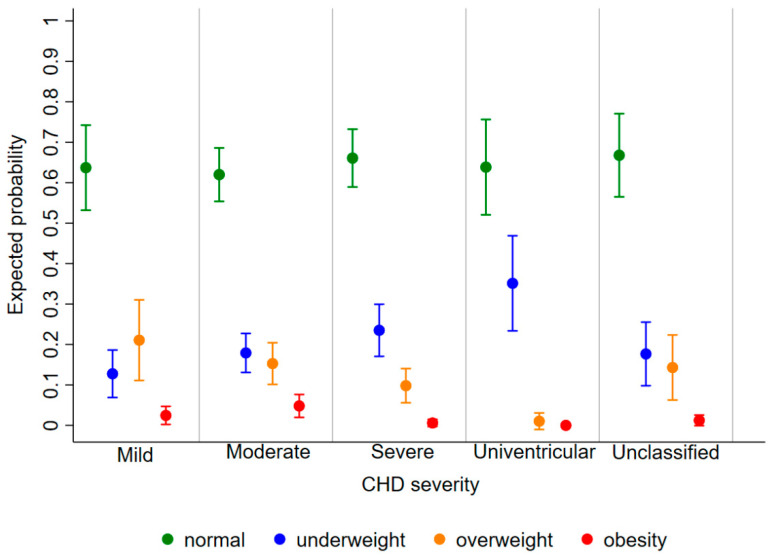
Expected probability of underweight (blue), normal weight (green), overweight (orange), and obesity (red) according to CHD severity.

**Figure 5 children-09-00884-f005:**
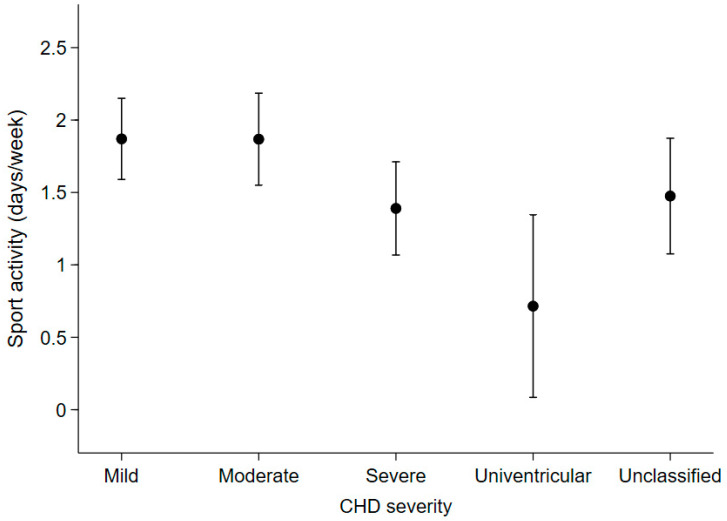
Expected number of days spent per week on sports activities in the last year according to disease severity in the 266 patients who were contactable by telephone at the end of the study.

**Table 1 children-09-00884-t001:** Main characteristics of patients included in the analysis.

	N = 368
Sex (female), %	40.8
Age at first recording (years), median (range)	5 (2–14)
Age at last measurement (years), median (range)	10 (2–18)
Duration of follow-up (years), median (range)	5.2 (0.0–12.7)
Number of measurements per patient, median (range)	7 (1–21)
Patients with only one recording of height and weight, %	24.7
CHD severity, %	
Mild	38.9
Moderate	26.3
Severe	18.5
Univentricular	3.3
Not classified	13.0
Number of surgical interventions, median (range)	0 (0–4)
Height at first recording (centimeters), mean ± sd	110.5 ± 18.8
Height z-score at first recording, mean ± sd	0.1 ± 1.2
Weight at first recording (kilograms), mean ± sd	20.2 ± 9.6
Weight z-score at first recording, mean ± sd	−0.5 ± 1.2
Body mass index at first recording, mean ± sd	15.7 ± 2.5
Body mass index z-score at first recording, mean ± sd	−0.8 ± 1.9
BMI category at first recording, %	
Underweight	24.7
Normal weight	58.2
Overweight	15.2
Obese	1.9
Age at end of the study (years) *, median (range)	12 (7–18)
CHD severity *, %	
Mild	34.6
Moderate	28.2
Severe	20.3
Univentricular	2.6
Not classified	14.3
Number of days spent per week in sports activities *, mean ± sd	1.7 ± 1.3

CHD: congenital heart disease; sd: standard deviation; * of 266 patients who were reachable by phone at the end of the study.

**Table 2 children-09-00884-t002:** Associations of age and CHD severity with height, weight, and BMI z-scores.

		Height z-Score	Weight z-Score	BMI z-Score
	Variables of Interest	Beta (95%CI)	Beta (95%CI)	Beta (95%CI)
**Model 1 ***	Age (years)	0.00 (−0.01, 0.01)	0.03 (0.02, 0.05)	0.06 (0.03, 0.08)
**Model 2 ***	Age (years)	0.00 (−0.01, 0.02)	0.03 (0.01, 0.04)	0.05 (0.02, 0.08)
	Severity (vs. mild)			
	moderate	−0.13 (−0.49, 0.23)	−0.52 (−0.86, −0.17)	−0.88 (−1.41, −0.36)
	severe	−0.19 (−0.58, 0.20)	−0.70 (−1.08, −0.32)	−1.12 (−1.67, −0.57)
	univentricular	−1.13 (−1.81, −0.45)	−1.85 (−2.5, −1.16)	−1.73 (−2.60, −0.86)
	not classified	−0.07 (−0.41, 0.27)	−0.04 (−0.39, 0.32)	−0.12 (−0.51, 0.28)

CI: confidence interval. * Model 1: adjusted for sex. Model 2: adjusted for sex and the indicator of surgery/intervention (yes/no). The estimates in bold are statistically significant (*p*-value < 0.05).

**Table 3 children-09-00884-t003:** Comparison between patients with only one clinical visit (S1) and patients with at least two clinical visits (S2).

	S1 Patients N = 91	S2 Patients N = 277	*p* Value	Statistical Test
Sex (female), %	44.0	39.7	0.539	Fisher’s exact test
Age at first recording (years), median	3	6	<0.001	Mood’s Median test
CHD severity, %				
Mild	71.4	28.2	<0.001	Fisher’s exact test
Moderate	14.3	30.3		
Severe	4.4	23.1		
Univentricular	3.3	3.2		
Not classified	6.6	15.2		
BMI category at first Recording, %				
Underweight	18.7	26.7	0.269	Fisher’s exact test
Normal weight	67.0	55.2		
Overweight	13.2	15.9		
Obese	1.1	2.2		
Number of days spent per week in sports activities *, mean	1.8	1.7	0.468	Student’s *t*-Test

* Of 222 patients who were reachable by phone at the end of the study.

## Data Availability

All data are available upon request.

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
