# Peer review of "Congenital Heart Disease: Growth Evaluation and Sport Activity in a Paediatric Population"

_children, 2022, doi:10.3390/children9060884_

Round 1
Reviewer 1 Report
Literature in the introduction seems to be outdated - please consider more using more recent literature. Consider the following systematic review on the topic: DOI: 10.3390/ijerph18189931
The authors seem to mention when the children were diagnosed with CHD, but not the point in time at which the anthropometric data was collected - musty clarify!
"Information on patient’s sport activity was collected through a single telephone interview with their parents or guardians at the end of thestudy (October 2018)" - I certainly believe this to be a substantial flaw for two reasons:
1) subjectively estimated physical activity has shown to not be not the most accurate measure in patients with CHD - please check DOI: 10.1159/000519286 DOI: 10.1016/j.jpeds.2019.09.077 and DOI: 10.1016/j.ahj.2021.07.004
2) Why did you choose # of days a child exercised / participated in sport activities? How did you define exercise / sport activities? Did you in your question adhere to any national or international guideline on PA i.e. WHO - please clarify.
Generally, I suggest to remove part about sport activity in this manuscript completely because the data collection was not clean and as a subjective, retrospective measure it has been demonstrated to be inadequate. Especially when the authors where asking about recalling activity over an entire year. Just focusing on overweight and obesity will in my opinion improve your mansucript
As it comes to sports activity, it would be interesting to include the sports recommendation policy by the physicians at the institution the patients where recruited from - are you generally very liberal or very conservative in prescribing exercise to your patients. Do you do exercise testing such as CPET to clear your patients for sport? Please include
Reviewer 2 Report
The publication is rather general (especially in the part on physical activity), but it raises a very important and current issue. It presents worrying trends in overweight and obesity as well as limiting physical activity in children with congenital heart defects. An article written correctly qualifies for publication.
Reviewer 3 Report
The authors perform a retrospective review of their congenital heart disease patients to assess the trajectory of body habitus over time relative to the severity of their congenital heart disease. The authors also perform single assessment of current sports participation in their cohort to determine the current degree of physical activity in their cohort. Their findings suggest that patients with mild-moderate forms of CHD are likely to develop higher degrees of overweight/obesity than patients with more severe CHD. This is contrasted by the overall low degree of physical activity within this group, particularly within the single ventricle population.
The authors provide compelling data that further support the trend seen around the world in CHD patients in developed countries of declining habits to promote long-term cardiovascular health. The study was performed overall well, though their physical activity assessments were limited to subjective parent reports without objective assessments of exercise capacity. However the data warrants publication to highlight the current trend in overall health in our CHD population.
The authors should consider the following comments:
1. The authors demonstrate longitudinal growth within their cohort, but they only have single time point data for 25% of their cohort. This discrepancy should be noted in study limitations.
2. Do the authors perform exercise stress testing for their patients as routine follow-up? Are they able to include any objective measures of exercise capacity or changes over time to correlate with their weight changes?
3. Did the authors have adequate numbers to compare weight differences in patients who were more active vs less active?
4. The underweight phenomenon in single ventricle patients is well-known, likely correlated with an overall decrease in muscle mass. This in turn can affect their Fontan hemodynamics by affecting the muscular contribution to systemic venous return and pulmonary circulation. The authors may wish to highlight this difference as well, given there are an increasing number of studies looking at the role of resistance exercise in the Fontan population.
5. The authors may also wish to include some of the data regarding physical activity programs in the CHD population, as there is an increasing body of literature highlighting the role of physical activity/cardiac rehab in this group and its impacts on exercise capacity and quality of life.
